# Bridging the Gap: Sketch-Aware Interpolation Network for High-Quality Animation Sketch Inbetweening

## ABSTRACT

Hand-drawn 2D animation workflow is typically initiated with the creation of sketch keyframes. Subsequent manual inbetweens are crafted for smoothness, which is a labor-intensive process and the prospect of automatic animation sketch interpolation has become highly appealing. Yet, common frame interpolation methods are generally hindered by two key issues: 1) limited texture and colour details in sketches, and 2) exaggerated alterations between two sketch keyframes. To overcome these issues, we propose a novel deep learning method - Sketch-Aware Interpolation Network (SAIN). This approach incorporates multi-level guidance that formulates region-level correspondence, stroke-level correspondence and pixel-level dynamics. A multi-stream U-Transformer is then devised to characterize sketch inbetweening patterns using these multi-level guides through the integration of self / cross-attention mechanisms. Additionally, to facilitate future research on animation sketch inbetweening, we constructed a large-scale dataset - STD-12K, comprising 30 sketch animation series in diverse artistic styles. Comprehensive experiments on this dataset convincingly show that our proposed SAIN surpasses the state-of-the-art interpolation methods. Our code and dataset will be publicly available.

## CCS CONCEPTS

• **Computing methodologies → Computer vision problems**.

## KEYWORDS

Sketch Interpolation, Hand-drawn Traditional Animation, Dataset STD-12K, Multi-level Correspondence, Multi-stream Transformer

## 1 INTRODUCTION

Hand-drawn 2D animation is extensively used in the animation industry for unique artistic expression, emotional depth and versatility. Notably, Your Name (2016) and Big Fish & Begonia (2016) achieved enormous success in recent years. The hand-drawn 2D animation workflow typically involves three key stages: sketching keyframes, inbetweening keyframes to produce intermediate sketch frames (i.e., inbetweens), and colorization to produce the final, full-color animations. The meticulous creation of inbetweens is crucial for achieving a smooth animation with lifelike motion transitions, effectively conveying the intended story or message. For a feature-length animation created through this process, the

**Unpublished working draft. Not for distribution.**

Permission to make digital or hard copies of all or part of this work for personal or classroom use is granted without fee provided that copies are not made or distributed for profit or commercial advantage and that copies bear this notice and the full citation on the first page. Copyrights for components of this work owned by others than the author(s) must be honored. Abstracting with credit is permitted. To copy otherwise, or republish, to post on servers or to redistribute to lists, requires prior specific permission and/or a fee. Request permissions from permissions@acm.org.

*ACM MM, 2024, Melbourne, Australia*

© 2024 Copyright held by the owner/author(s). Publication rights licensed to ACM.
ACM ISBN 978-x-xxxx-xxxx-x/YY/MM
https://doi.org/10.1145/nnnnnnn.nnnnnnn

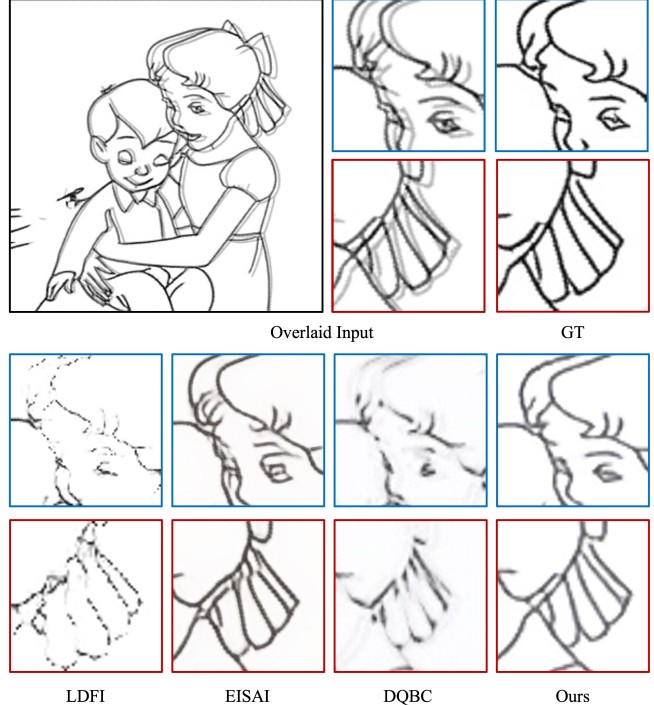

**Figure 1: Illustration of sketch inbetweening: the limitations of relevant methods in comparison to our SAIN, including LDFI [17] for sketches, EISAI [3] for color animations, and DQBC [40] for videos.**

sheer volume of required inbetweens can be staggering [33], making it a highly specialized and labor-intensive task and serving as a limiting factor in overall animation productivity.

To streamline the process of 2D sketch animation production, various studies have focused on the automatic synthesis of inbetweening sketch frames, which take two consecutive sketch keyframes as input and produce interpolated intermediate sketch frames (i.e., inbetweens) as output. These methods can be categorised into stroke-based and image-based. The stroke-based methods often rely on a labor-intensive pre-processing step for sketch vectorisation [29, 35, 38], whilst subpar vectorisation quality can negatively impact the final outcomes. Image-based methods treat sketch frames as bitmap images, applying conventional image or video interpolation algorithms. However, they commonly face two significant challenges: 1) the absence of texture and color details in sketch frames, hindering reliable image-based inbetweening correspondence, and 2) exaggerated changes due to substantial object movements between two consecutive sketch keyframes [17]. As a result, when image-based methods, especially devised for videos

[40] and colour animations [3, 30], are applied to sketch interpolation, they invariably introduce various artifacts into the produced interpolated frames. These discrepancies can adversely affect the continuity and quality of the animation produced. As shown in Figure 1, LDFI [17] proposed for sketch interpolation generates broken strokes due to the missing sketch keypoint correspondence, while EISAI [3] proposed for interpolating color animation frames and DQBC [40] for video interpolation introduce blurriness (ornaments) and artifacts (e.g., distortion in face regions).

Therefore, in this study, we propose a novel deep learning method for sketch interpolation, the Sketch-Aware Interpolation Network (SAIN), to comprehend and model the intricate and sparse patterns found in hand-drawn animation sketches. SAIN adopts a sketch-aware approach that integrates multi-level sketch-related guidance through three distinct aspects: 1) *pixel-level dynamics* at a fine level with a bi-directional optical flow estimation module, 2) *stroke-level correspondence* with a stroke matching and tracking mechanism for obtaining stroke keypoint traces and 3) *region-level correspondence* at a coarse level with a region matching and bi-directional optical flow aggregation module. The usage of term "stroke" here align with the stroke-based methods, referring to every single lines or outlines in a sketch keyframe. Guided by these multi-level perspectives, a multi-stream U-Transformer architecture is further devised to produce the intermediate sketch frames. It consists of two attention-based building blocks: *convolution and self-attention block* (CSB) and the *convolution and cross-attention block* (CCB) to leverage the diverse multi-level insights for producing precise inbetween sketch patterns. To facilitate the research on hand-drawn animation sketch inbetweening, we constructed a large-scale sketch triplet dataset: STD-12K, from 30 sketch animation series over 25 hours with various artistic styles. Comprehensive experiments demonstrate that our SAIN clearly outperforms the state-of-the-arts for animation sketch interpolation.

Overall, the key contributions of this study are as follows:

- A novel deep learning architecture, SAIN, for sketch interpolation by effectively formulating sparse sketch patterns with sketch-aware multi-level guidance.
- A novel self / cross-attention based multi-stream U-Transformer design with the multi-level guidance.
- A large-scale sketch triplet dataset with various artistic styles constructed for the research community.

## 2 RELATED WORK

Frame interpolation aims to synthesise the intermediate frames between a pair of given frames. It can be categorised into three major approaches regarding frame contents: sketch interpolation [11, 17, 32, 35, 37], video interpolation [2, 16, 18, 20, 21], and animation interpolation [25, 30].

### 2.1 Sketch Interpolation

Sketch interpolation is to produce raw animated sketch frames to streamline the 3-stage process of 2D sketch animation. Generally, existing studies can be categorised into stroke-based [11, 35, 37] and image-based approaches [17, 32].

Stroke-based methods for hand-drawn frames generally start with stroke vectorisation, and then perform stroke deformation [35]

or construct specific structure units with vertices [37]. Recently, the transformer-based methods Sketchformer [22] and AnimeInbet [29] were introduced, capitalizing on the success of this architectural approach in various computer vision tasks. However, the preliminary processing heavily relies on extra software or techniques, and human animators often need to fine-tune the results of vectorization to ensure a smooth and accurate portrayal of intended movements. This introduces additional complexity, and substandard vectorization quality can adversely affect the inbetweening process, which makes stroke-based methods challenging to apply into the animation workflow. While interactive matching algorithms have been investigated for this process in [37], scalability issues become prominent when the number of strokes in frames increases with subpar quality of hand-drawns.

Image-based methods were initially studied in [32], where an as-rigid-as image registration and an interpolation scheme were introduced to bypass the vectorization phase inherent in stroke-based methods, demonstrating its potential in dealing with intricate stroke patterns. Recently, optical flow of sketches has been adopted to characterize the motions of characters and objects within an animation. In LDFI [17], a distance transform mechanism was adopted, which engages the intensity gradients of the sketches with an enhances optical flow estimation. However, this mechanism can compromise sketch details, particularly in complex scenarios where strokes undergo significant changes.

### 2.2 Animation Interpolation

Different from natural videos in the real-world, cartoon animations mainly consist of expressive strokes and colour pieces. They often contain various non-linear and exaggerated motions. SGCVI [13] allowed users to generate inbetween frames guided by one user-input sketch. AnimeInterp [30] introduced a segment-guided matching module to estimate the optical flow for different colour pieces separately and a recurrent prediction module to address non-linear motions. EISAI [3] was recently proposed with a forward-warping interpolation architecture SoftsplatLite and a distance transform module to improve the perceptual quality.

### 2.3 Video Interpolation

Early video interpolation studies were based on the optical flows between two input frames to represent and formulate motion patterns, exemplified by methods using a bidirectional optical flow method, such as in [6]. With the success of deep learning techniques in diverse computer vision tasks, kernel-based methods integrated convolution neural networks (CNNs) [12, 19] for efficient motion estimation and generation. Recently, due to the great success of visual transformers [5, 14], transformer-based methods with self-attentions have been studied for video interpolation, adaptively addressing long-range pixel dependencies. Self-attentions were utilized to formulate the representation of each input frame in [26]. Cross-attentions between the input frame pairs were further studied in [9]. Optical flows were introduced to the transformer modelling as well to assist the formulation of motion dynamics [16]. DQBC [40] followed the flow-based paradigm and integrated correlation modeling to enhance the flow estimation. Yet, it is challenging to obtain reliable flow information between two animation sketches.

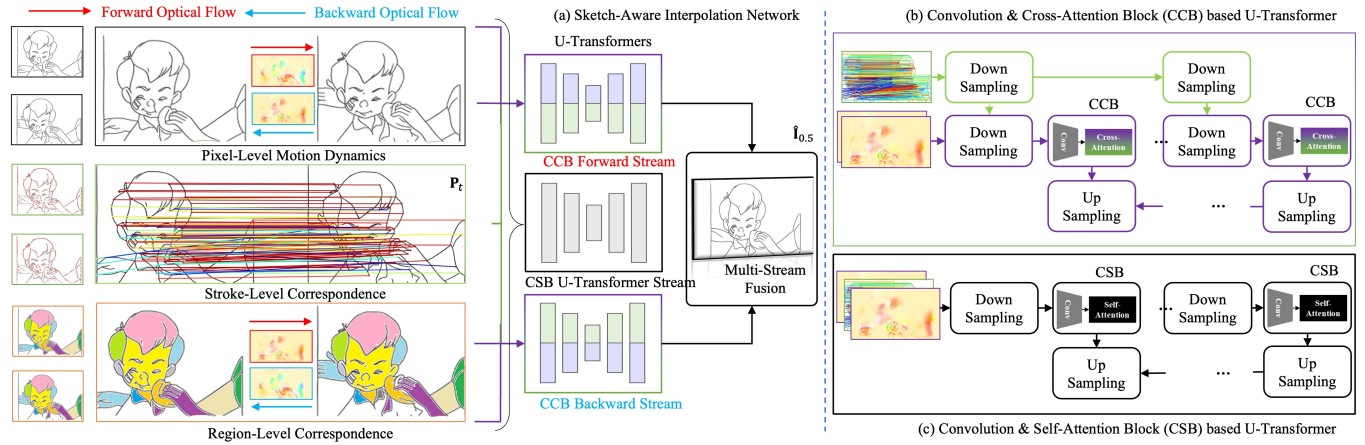

**Figure 2: Illustration of the proposed Sketch-Aware Interpolation Network (SAIN).**

## 3 METHODOLOGY

As shown in Figure 2, SAIN takes two consecutive sketch keyframes as input and outputs an interpolated intermediate frame. It involves region, stroke, and pixel-level guidance to accurately capture and recognize the sparse characteristics of sketch animations. Correspondingly, a multi-stream U-Transformer is devised with self-attention and cross-attention based building blocks to produce intermediate sketches in a multi-scale manner. In this section, we first explain the problem formulation, then the details of SAIN's key components. To clarify and avoid ambiguity, it should be noted that the term *stroke* refers to each individual pen or brush movement that contributes to the creation of the complete sketch keyframes during drawing.

### 3.1 Problem Formulation

Denote two consecutive animation sketch keyframes as $I_0, I_1 \in \mathbb{R}^{H \times W \times C}$, where $H$, $W$, and $C$ denote their height, width and the number of channels, respectively. Animation sketch interpolation takes the two sketch keyframes to estimate an interpolated sketch frame $\hat{I}_{\bar{t}} \in \mathbb{R}^{H \times W \times C}$ for the ground truth $I_{\bar{t}} \in \mathbb{R}^{H \times W \times C}$, ($0 < t < 1$), where $\bar{t} = 0.5$ for an intermediate frame between the keyframes by following the existing practices in the literature [16, 17].

### 3.2 Pixel-Level Motion Dynamics

To capture pixel-level motion dynamics, optical flows are estimated between the keyframes $I_0$ and $I_1$. This allows us to output refined sketch keyframes, denoted as $\dot{I}_0$ and $\dot{I}_1$, that incorporate these motion dynamics. Pixel-level dynamics are good at characterizing the patterns of all pixels, and they can also maintain the patterns that might be missed in the subsequent sketch- and region-level pattern formulation. Specifically, given a target timestamp $\bar{t}$, an optical flow estimator [16] predicts the bi-directional flows: $O_{\bar{t} \to 0}$ and $O_{\bar{t} \to 1}$. Next, the refined keyframes with pixel-level motion dynamics can be obtained as:

$$\dot{I}_0 = \mathcal{W}(I_0, O_{\bar{t} \to 0}), \; \dot{I}_1 = \mathcal{W}(I_1, O_{\bar{t} \to 1}), \tag{1}$$

where $\mathcal{W}$ is an image warping function [7] to fuse the two inputs with a pre-defined sampling strategy.

### 3.3 Stroke-Level Correspondence

To characterise sketch-based motion patterns for interpolation, stroke-level correspondence is formulated between sketch keyframes. Specifically, a stroke keypoint matching and tracking mechanism is devised for this purpose, which assists in a cross-frame stroke understanding.

**Point-wise matching** aims to produce a set of matched salient points between the input strokes in $I_0$ and $I_1$, encompassing a *salient point identification* step for individual keyframes and a *salient point matching* step between the paired keyframes. First, given a sketch frame, the stroke salient points can be identified with their feature descriptors that characterise their local point-wise patterns, which can be formulated by algorithms such as SuperPoint [4]. We denote the identified salient points as $p_i = (x_i, y_i, c_i)$ and their visual descriptor as $d_i \in \mathbb{R}^{D_p}$ for the $i^{\text{th}}$ detected stroke point. In detail, $x_i$ and $y_i$ are the coordinates of the $i^{\text{th}}$ salient point, and $c_i$ indicates the detection confidence. To this end, given the two keyframes, the stroke salient point detection finds $N_s^{I_0}$ and $N_s^{I_1}$ points with their local features as: $p_i^{I_0}$ and $d_i^{I_0}$, $i = 1, ..., N_s^{I_0}$ for $I_0$, and $p_j^{I_1}$ and $d_j^{I_1}$, $j = 1, ..., N_s^{I_1}$ for $I_1$, respectively.

Salient point matching further establishes the point correspondence between paired keyframes in line with their feature descriptors and obtains a set of point pairs with confidence scores. Mathematically, we have their confidence as:

$$c_{ij} = \mathcal{U}(p_i^{I_0}, p_j^{I_1}, d_i^{I_0}, d_j^{I_1}), \forall i, j, \tag{2}$$

where $\mathcal{U}$ is a function to evaluate the confidence with the point coordinates and descriptors. A SuperGlue [24] method is adopted for this purpose. As shown in Figure 2, colorful line connections indicate the matched point pairs with high confidence scores over a threshold $\theta$, where their colours indicate the magnitude of the confidence scores: red for a higher score and blue for a lower score.

**Point-wise tracking.** For a brief interval between $\mathbf{I}_0$ and $\mathbf{I}_1$, the movements are assumed to follow a linear path with respect to a temporal indicator $t$, $t \in [0, 1]$. Mathematically, consider a matched pair of salient points, such as the $i^{\text{th}}$ keypoint in $\mathbf{I}_0$ and the $j^{\text{th}}$ keypoint in $\mathbf{I}_1$; we have:

$$\mathbf{p}_{ij}^{I_t} = t \times \mathbf{p}_i^{I_0} + (1 - t) \times \mathbf{p}_j^{I_1}, \tag{3}$$

where $\mathbf{p}_{ij}^{I_t}$ is an estimation of the intermediate trace of the stroke salient point at time $t$.

To this end, the traces of these salient points at time $t$ can be conceptualized as a 2D frame, characterized by their coordinates. We represent the stroke-level correspondence using this trace, denoted as $\mathbf{P}_t \in \mathbb{R}^{H \times W \times C}$, $t \in [0, 1]$. This trace shares the same dimension as $\mathbf{I}_0$ and $\mathbf{I}_1$.

### 3.4 Region-Level Correspondence

Sketch frames generally contain clear outline strokes and enclosed areas. To leverage this regional nature for interpolation, region-correspondence is constructed between two sketch keyframes. Specifically, regions can be identified as segmentation maps using methods such as the trapped-ball algorithm. For $\mathbf{I}_0$ and $\mathbf{I}_1$, $N_r^{\mathbf{I}_0}$ and $N_r^{\mathbf{I}_1}$ regions are identified, respectively. Pre-trained CNN features for these maps can be formulated, allowing the pixel-based features within a region to be pooled as a $D_r$-dimensional vector, thereby characterizing each region. Given the coordinates and features of these regions, a match can be established between the regions across keyframes akin to stroke correspondence.

For a region pair $(i, j)$, where $i$ indicates the $i^{\text{th}}$ region in $\mathbf{I}_0$ and $j$ indicates the $j^{\text{th}}$ region in $\mathbf{I}_1$, bi-directional optical flows can be estimated based on their features as $\mathbf{f}_{\bar{t} \to 1}(i, j)$ and $\mathbf{f}_{\bar{t} \to 0}(j, i)$. By summing up all regional optical flows, we have $\mathbf{F}_{\bar{t} \to 1} = \sum_{(i,j)} \mathbf{f}_{\bar{t} \to 1}(i, j)$ and $\mathbf{F}_{\bar{t} \to 0} = \sum_{(j,i)} \mathbf{f}_{\bar{t} \to 0}(j, i)$. To this end, the keyframes are refined with region-level correspondence information as follows:

$$\ddot{\mathbf{I}}_0 = \mathcal{W}(\mathbf{I}_0, \mathbf{F}_{\bar{t} \to 1}), \quad \ddot{\mathbf{I}}_1 = \mathcal{W}(\mathbf{I}_1, \mathbf{F}_{\bar{t} \to 0}), \tag{4}$$

where the $\mathcal{W}$ is an image warping function.

### 3.5 Multi-Stream U-Transformers

A multi-stream U-Transformer is devised to characterise the inbetweening patterns by jointly considering region, stroke and pixel-level dynamics. These streams are based on two building blocks: convolution and self-attention block (CSB) and convolution and cross-attention block (CCB).

**CSB U-Transformer stream.** This stream fully adopts the multi-level dynamics for the motion patterns between $\mathbf{I}_0$ and $\mathbf{I}_1$. Specifically, a concatenation is conducted on patterns regarding pixel-level ($\dot{\mathbf{I}}_0, \dot{\mathbf{I}}_1$), stroke-level ($\mathbf{P}_t$) and region-level ($\ddot{\mathbf{I}}_0, \ddot{\mathbf{I}}_1$), followed by a number of convolution layers to obtain a coarse-level intermediate sketch representation $\mathbf{X}_{\text{coarse}}$. To further formulate the inbetweening patterns from a fine-level perspective, CSB with self-attentions [34] is introduced to construct a U-Net [23] like a stream with an encoder-decoder architecture.

The encoder consists of a series of CSBs. Specifically, a CSB consists of a convolution layer for modelling local sketch patterns and a multi-head self-attention for a global modelling purpose. In pursuit of an overall encoder pyramid structure of $S$ scales, a downsampling operator is introduced with CSB to formulate a feature map at its corresponding scale. For the $s^{\text{th}}$ CSB, which is for the $s^{\text{th}}$ scale, $s = 0, ..., S - 1$, its output feature map is obtained as:

$$\mathbf{X}_{s+1}^{\text{CSB}} = \text{CSB}_s(\mathbf{X}_s^{\text{CSB}}), \tag{5}$$

where $\mathbf{X}_{s+1}^{\text{CSB}} \in \mathbb{R}^{H_{s+1}^{\text{CSB}} \times W_{s+1}^{\text{CSB}} \times C_{s+1}^{\text{CSB}}}$. Particularly, the first CSB takes $\mathbf{X}_0^{\text{CSB}} = \mathbf{X}_{\text{coarse}}$ as its input.

In detail, $\mathbf{X}_s^{\text{CSB}}$ is first with a convolution layer for local modelling. Note that for notation simplicity we keep using $\mathbf{X}_s^{\text{CSB}}$ as the convolution-filtered results for the following discussion. Next, $\mathbf{X}_s^{\text{CSB}}$ is divided into $K_s^{\text{CSB}} = H_s^{\text{CSB}} W_s^{\text{CSB}} / M_s^2$ sub-patches of size $M_s \times M_s$ following the general practice of a visual transformer [15]. By treating the pixel-wise values within the $k^{\text{th}}$ patch as a representation vector $\mathbf{x}_{s,k}^{\text{CSB}} \in \mathbb{R}^{M_s^2 \times C_s^{\text{CSB}}}$, $\mathbf{X}_s^{\text{CSB}}$ can be viewed in a matrix form, where $\mathbf{X}_s^{\text{CSB}} = [\mathbf{x}_{s,1}^{\text{CSB}\top}, ..., \mathbf{x}_{s,K_s^{\text{CSB}}}^{\text{CSB}\quad\top}]$. Then, the matrices of Key, Query, and Value in a self-attention can be computed to obtain a frame-level sketch understanding:

$$\mathbf{Q}_s^{\text{CSB}} = \mathbf{X}_s^{\text{CSB}} \mathbf{W}_{Q_s}^{\text{CSB}},$$
$$\mathbf{K}_s^{\text{CSB}} = \mathbf{X}_s^{\text{CSB}} \mathbf{W}_{K_s}^{\text{CSB}},$$
$$\mathbf{V}_s^{\text{CSB}} = \mathbf{X}_s^{\text{CSB}} \mathbf{W}_{V_s}^{\text{CSB}}, \tag{6}$$

where $\mathbf{W}$ indicates a matrix with learnable parameters for a linear projection. Next, the attention can be computed as:

$$\mathbf{X}_{s+1}^{\text{CSB}} = \text{softmax}\left(\frac{\mathbf{Q}_s^{\text{CSB}} \mathbf{K}_s^{\text{CSB}\top}}{\sqrt{d_s^{\text{CSB}}}}\right) \mathbf{V}_s^{\text{CSB}}, \tag{7}$$

where $d_s^{\text{CSB}}$ is the dimension of queries, keys and values. Specifically, we denote $\mathbf{X}_{\text{fine}} = \mathbf{X}_S^{\text{CSB}}$ as a fine-level feature map. Note that this encoder structure, along with its CSB blocks, can work seamlessly with multi-head self-attentions, which enables the extension of frame-level sketch patterns from multiple perspectives. Furthermore, the structure is compatible with the Swin-based window strategies used during patch construction, keeping efficiency in consideration. Finally, the decoder with a number of deconvolution layers upsamples $\mathbf{X}_{\text{fine}}$ as a synthetic CSB frame feature map $\hat{\mathbf{I}}_{\bar{t}}^{\text{CSB}}$.

**CCB U-Transformer stream.** In the CSB stream, multi-level patterns contribute equally to the formulation of the feature map. Leveraging the stroke and region-based characteristics of sketch animes, CCB U-Transformer streams with an encoder-decoder structure are devised to provide diversified modelling perspectives for sketch interpolation.

For its encoder, similar to the CSB stream, a series of CCBs are adopted for different pyramid scales $s = 0, ..., S - 1$. For the $s^{\text{th}}$ CCB, a feature map is formulated as:

$$\mathbf{X}_{s+1}^{\text{CCB}} = \text{CCB}_s(\mathbf{X}_s^{\text{CCB}}, \mathbf{Y}_s^{\text{CCB}}), \tag{8}$$

where $\mathbf{X}_s^{\text{CCB}}, \mathbf{Y}_s^{\text{CCB}} \in \mathbb{R}^{H_s^{\text{CCB}} \times W_s^{\text{CCB}} \times C_s^{\text{CCB}}}$. Specifically, stroke-level patterns $\mathbf{P}_t$ are concatenated and downsampled with learnable convolutions as $\mathbf{Y}_s^{\text{CCB}}$ for key and value computations in the cross-attention. In terms of query, pixel-level dynamics and region-level correspondence are utilized. Since they are bi-directional, dual CCB U-Transformer streams are introduced for concatenated query features $\dot{\mathbf{I}}_0$ and $\mathbf{I}_0$, or $\dot{\mathbf{I}}_1$ and $\mathbf{I}_1$. In particular, the first CCB takes

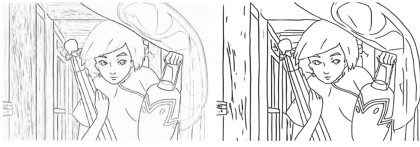

(a) Animation frame  (b) Coarse Strokes  (c) Refined Strokes

**Figure 3: Illustration of the key steps in the pipeline constructing STD-12K from an animation in color.**

$\mathbf{F}_0^{CCB} = Conv(\dot{\mathbf{I}}_0, \mathbf{I}_0)$ or $Conv(\dot{\mathbf{I}}_1, \mathbf{I}_1)$ as its inputs, where $Conv$ is a function for concatenation and downsampling, $(\mathbf{I}_0, \mathbf{I}_0)$ and $(\dot{\mathbf{I}}_1, \mathbf{I}_1)$ can be viewed as the forward and backward streams.

Finally, decoders produce synthetic intermediate feature maps: $\hat{\mathbf{I}}_{\bar{t},0}^{CCB}$ or $\hat{\mathbf{I}}_{\bar{t},1}^{CCB}$ are obtained from the two streams regarding the inputs $\mathbf{X}_0^{CCB} = \dot{\mathbf{I}}_0$ and $\mathbf{X}_0^{CCB} = \dot{\mathbf{I}}_1$, respectively.

**Multi-stream fusion.** Upon obtaining the intermediate feature maps $\hat{\mathbf{I}}_{\bar{t}}^{CSB}$, $\hat{\mathbf{I}}_{\bar{t},0}^{CCB}$ and $\hat{\mathbf{I}}_{\bar{t},1}^{CCB}$ from the CSB and CCB streams, a fusion mechanism then yields the final estimation $\hat{\mathbf{I}}_{\bar{t}}$ of the interpolated frame $\mathbf{I}_{\bar{t}}$, by which all feature maps are concatenated and go through a series of convolutions.

### 3.6 Training Loss

We denote the computations of the proposed method as a function $f$ with learnable weights $\Theta$. To obtain $\Theta$, $\ell_1$ reconstruction based loss is adopted to optimise pixel-wise difference between the ground truth sketch frame $\mathbf{I}_t$ and the interpolated frame $\hat{\mathbf{I}}_t$. Mathematically, we have:

$$\mathcal{L}_1 = ||\hat{\mathbf{I}}_{\bar{t}} - \mathbf{I}_{\bar{t}}||_1, \hat{\mathbf{I}}_{\bar{t}} = f(\mathbf{I}_0, \mathbf{I}_1 | \Theta). \tag{9}$$

To further improve the synthesized details, we apply a perceptual LPIPS loss [39], denoted as $\mathcal{L}_{\text{lpips}}$. Jointly, the proposed model is optimized as:

$$\arg\min_{\Theta} \mathcal{L} = \lambda_1 \mathcal{L}_1 + \lambda_{\text{lpips}} \mathcal{L}_{\text{lpips}}. \tag{10}$$

## 4 EXPERIMENTS & DISCUSSIONS

### 4.1 Dataset

Due to the lack of publicly available datasets for animation sketch interpolation, by following the protocols of existing video interpolation datasets such as Vimoe-90K [36] and UCF101 [31], we constructed an animation sketch dataset based on ATD-12K [30], namely Sketch Triplet Dataset-12K (STD-12K), for evaluation and facilitating the research on this topic. STD-12K is a large-scale sketch triplet dataset extracted from 30 animation movies with extensive artistic styles. To convert an animation frame to a sketch frame, Sketch Keras was first used to detect and extract the contours in a frame and rough strokes can be obtained. Next, a sketch simplification procedure [27, 28] was introduced to remove blurry and trivial strokes, which also refined the basic and necessary sketch lines. Figure 3 depicts the key steps to construct this dataset.

### 4.2 Implementation Details

**Network architecture.** For $\mathbf{P}_t$ in stroke matching and tracking, we specified a stroke correspondence sequence with $t = 0.5$. An

appropriate setting to use these temporal information would have impact on the interpolation accuracy as indicated in the experiments. For the optical flow computations, we first predicted the coarse flows using convolutional flow prediction network and then refined the coarse flows in a coarse-to-fine manner following an existing practice as in VFIformer [16]. For the CSB component, a swin-based strategy was adopted with a window size of $8 \times 8$. The number of channels in its convolution layer was set to 24. The CCB component was with the same setting as CSB. Each U-Net like transformer stream contained 3 CSBs or CCBs.

**Training details.** The proposed method was trained using an AdamMax optimizer [10] with $\beta_1 = 0.9$ and $\beta_2 = 0.999$. The weights of loss terms were set to $\lambda_1 = 70$ and $\lambda_{\text{lpips}} = 30$. The training batch size was set to 4. The SAIN was trained for 50 epochs with a learning rate that initially was set as $2e^{-4}$ and a weight decaying factor was set to $1e^{-4}$. The sketch frames were resized and cropped into a resolution of $384 \times 192$, and they were also augmented with a random flipping operator. It took approximately 72 hours on an NVIDIA A6000 GPU for the training procedure.

### 4.3 Overall Performance

**Evaluation metrics.** For quantitative evaluation, we adopt commonly used visual quality assessment metrics: Peak Signal-to-Noise Ratio (PSNR), Structural Similarity Index Measure (SSIM) scores, Interpolation Error (IE) [1] which measures pixel-wise difference between the interpolated and ground-truth sketches, and Chamfer Distance (CD) which measures the dissimilarity between two sets of points. For readability, IE is scaled by $1e2$ and CD by $1e4$.

**Methods for comparisons.** SAIN is compared with the recent state-of-the-art methods, encompassing stroke-based sketch frame interpolation method AnimeInbet [29], Sketchformer [22] and image-based LDFI [17], animation-based methods SGCVI [13] and EISAI [3], and video-based methods: Super SloMo [8], AdaCof [12], SoftSplat [18], RIFE [7], VFIT [26], VFIformer [16] and DQBC [40]. All the SOTA models were retrained on our proposed dataset, STD-12K, with the exception of AnimeInbet, for which we utilized the authors' pre-trained model. The decision to use the pre-trained AnimeInbet model stemmed from the challenges we encountered in applying the authors' recommended method for vectorizing our sparse images. This process spans over a month for preprocessing our training dataset and it creates unsatisfactory quality sketches that are not suitable for further training.

**Quantitative evaluation.** As shown in Table 1, our SAIN consistently outperforms the other methods with PSNR 20.32, SSIM 0.8727, IE 10.09 and CD 1.54. SAIN successfully addresses the sparsity nature of sketch animations, as evidenced by the lowest CD score which indicates a high degree of similarity between the interpolated sketch and the ground truth, given the sketch as a set of points. Video-based methods generally underperform when compared to the animation-based approach - EISAI. The only exception is VFIformer, which slightly surpasses EISAI in terms of the IE metric. This underscores the challenges that video-based methods face when dealing with the sparse patterns intrinsic to animations. When comparing our proposed SAIN to EISAI, it becomes evident that proper sketch-based mechanisms are essential, especially given that sketch-based animations often lack color and detailed texture

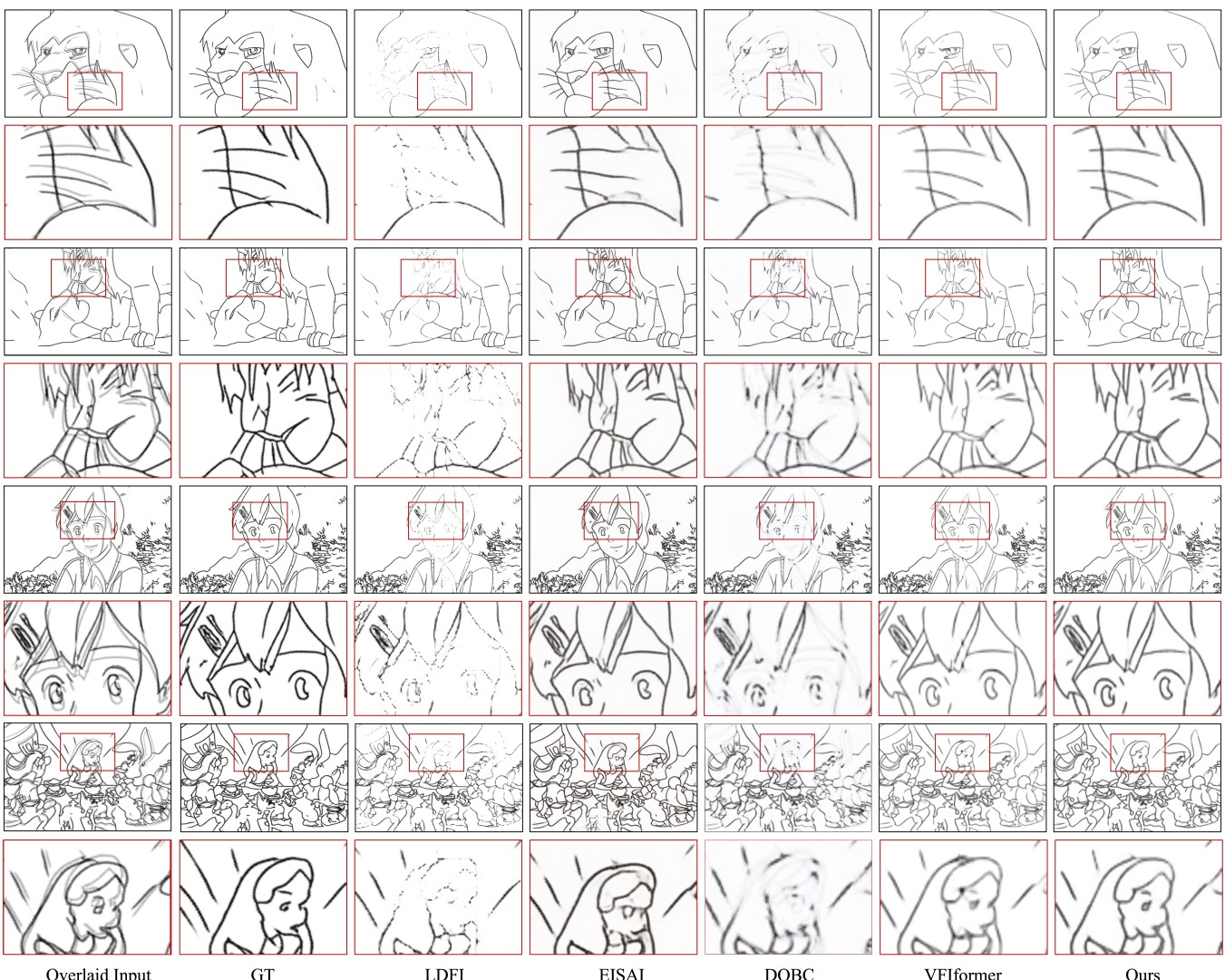

|  Overlaid Input | GT | LDFI | EISAI | DQBC | VFIformer | Ours |

**Figure 4: Qualitative comparison between the proposed SAIN (Ours) and the state-of-the-art interpolation methods.**

structures. However, while AdaCoF performs well in terms of CD, its PSNR is poor, suggesting that it prioritizes rough sparse structures and ignores detailed patterns. Other video-based methods perform even worse in terms of CD metrics. EISAI, an animation-based method, achieves comparable PSNR and SSIM scores, but still struggles with sparsity, particularly without texture and color information. Finally, when compared to the sketch-based LDFI method, its intensity gradient mechanism results in inaccurate results, especially for complex scenarios with dramatic changes of strokes.

**Qualitative evaluation.** Figure 4 illustrates interpolation examples from simple to complex scenarios for the qualitative comparisons among different methods. The frames with a black border are full sketch frames, and we zoom in a specified region within a red window to observe the detailed interpolated patterns. Overall, it can

be observed that SAIN is capable to generate high-quality inbetweens, and the results produced by other methods generally have different type of artifacts such as blurriness and distortions. The first example is with the simplest strokes. The results produced by LDFI and DQBC missed many strokes due to the limitation in exploring sketch correspondence for alignments. While EISAI and VFIformer achieve improved performance, the issue of blurriness persists due to the lack of texture and color reference. With the increasing sketch complexity, these artifacts become more significant and the contents tend to be unrecognizable (e.g., the 3[th] example with DQBC) and distortion (e.g., suspecting angry face with EISAI for the 3[th] example). Note that in the zoom in region, AdaCof and LDFI failed to output contiguous strokes since insufficient correspondence information was extracted which result in missing

| Method (Year) | PSNR ↑ | SSIM ↑ | IE ↓ | CD ↓ |
|---|---|---|---|---|
| AnimeInbet (2023) | 12.30 | 0.5796 | 25.00 | 62.20 |
| Sketchformer (2020) | 17.23 | 0.7847 | 14.14 | 10.34 |
| LDFI (2019) | 18.18 | 0.8048 | 12.71 | 4.05 |
| SGCVI (2021) | 17.56 | 0.7850 | 13.56 | 3.68 |
| EISAI (2022) | _19.07_ | _0.8422_ | 11.62 | _1.76_ |
| Super SloMo (2018) | 18.05 | 0.7995 | 12.86 | 3.82 |
| AdaCoF (2020) | 18.08 | 0.8027 | 12.82 | 4.39 |
| SoftSplat (2020) | 17.08 | 0.7328 | 14.17 | 5.61 |
| VFIT (2022) | 8.45 | 0.5622 | 39.03 | 13.59 |
| RIFE (2022) | 15.11 | 0.6258 | 18.37 | 641.58 |
| VFIformer (2022) | 19.05 | 0.8387 | _11.59_ | 6.54 |
| DQBC (2023) | 18.60 | 0.8015 | 12.12 | 2.39 |
| **SAIN (Ours)** | **20.32** | **0.8727** | **10.09** | **1.54** |

**Table 1: Quantitative comparison between SAIN and the state-of-the-art interpolation methods.**

| Method | PSNR ↑ | SSIM ↑ | IE ↓ | CD ↓ |
|---|---|---|---|---|
| **SAIN** | **20.32** | **0.8727** | **10.09** | _1.54_ |
| w/o region corr. | _20.25_ | _0.8679_ | _10.19_ | **1.52** |
| w/o sketch corr. | 20.16 | 0.8586 | 10.29 | 2.07 |
| w/o CSB stream | 20.09 | 0.8620 | 10.36 | 1.85 |
| w/o CCB stream | 19.95 | 0.8628 | 10.46 | 1.67 |
| w/o pixel dynamics | 19.14 | 0.8424 | 11.43 | 1.80 |

**Table 2: Ablation studies on SAIN.**

sketch keypoints. EISAI and VFIformer generated either blurriness or phantom strokes due to the lack of feature correspondence between the sketch keyframes.

## 4.4 Ablation Study

Ablation studies were conducted to demonstrate the effectiveness of individual mechanisms in SAIN: from five aspects: stroke-level correspondence module, pixel-wise dynamic formulation, region-level correspondence module, convolution & cross-attention block based, and convolution & self-attention block based multi-stream transformer. To evaluate the contribution of each aspect, we remove one of such mechanisms in each experiment, and trained and evaluated the corresponding model on the STD-12K dataset.

**Pixel-wise dynamics**. By removing the pixel-wise dynamics, the CCB and CSB transformer streams take region-level correspondence $\ddot{\mathbf{I}}_0$ and $\ddot{\mathbf{I}}_1$, and stroke-level correspondence $\mathbf{P}'_t$ as inputs. As shown in Table 3, the absence of pixel-level motion information resulted in a deteriorated performance. Moreover, as shown in Figure 5, the interpolation examples exhibits blurring results, potentially due to the uncertain direction of pixel-level motion.

**Stroke-level correspondence**. The stroke-level correspondence $\mathbf{P}_t$ was removed from SAIN by excluding the CCB based transformer streams and $\mathbf{P}_t$ in the CSB based transformer stream. Only the four refined sketch frames $\dot{\mathbf{I}}_0$, $\dot{\mathbf{I}}_1$, $\ddot{\mathbf{I}}_0$ and $\ddot{\mathbf{I}}_1$ were adopted and fused

| Tracking strategy | PSNR ↑ | SSIM ↑ | IE ↓ | CD ↓ |
|---|---|---|---|---|
| 1/4, 1/2, 3/4 | 19.92 | 0.8643 | 10.52 | 1.50 |
| 1/3, 2/3 | 20.01 | 0.8596 | 10.46 | 1.67 |
| 1/2, 3/4 | **20.33** | 0.8686 | 10.07 | 1.55 |
| 1/4, 1/2 | 20.31 | _0.8694_ | _10.08_ | _1.54_ |
| **1/2** | _20.32_ | **0.8727** | **10.09** | **1.54** |

**Table 3: SAIN with different tracking strategies.**

in a single CSB transformer stream. The absence of stroke-level correspondence resulted in lower performance compared to full SAIN, which indicates the necessity of exploiting stroke patterns. By zooming out details shown in Figure 5 (row with blue outlines), SAIN without stroke correspondence has lower contrast and the black lines are less noticeable, which indicates SAIN without stroke correspondence have less confidence on outputs.

**Convolution cross-attention block**. Without CCB for a sketch focused modelling, it can be observed that the quantitative results shown in Table 5 are worse than those of the full SAIN. Specifically, for the second example in Figure 5, the intricate details of the clown's face is difficult to discern when CCB is not utilised. Moreover, CCB facilitates a robust learning with stroke correspondence, whilst an improper stroke guidance usage may result in inferior interpolation results for some scenarios.

**Convolution self-attention block**. When the CSB block is omitted, the quantitative results, as depicted in Table 5, demonstrate inferior performance compared to the full SAIN. In particular, in the first example (inside the princess's hair) illustrated in Figure 5, blurriness occured in the princess's hair when interpolation without CSB.

**Region-level correspondence**. Similar to the removal of pixel-wise dynamics, by removing the region-level correspondence, we instead take the refined outputs from pixel-wise dynamics module $\dot{\mathbf{I}}_0$ and $\dot{\mathbf{I}}_1$ to explore another situation not fully utilizing the correspondence. SAIN without region correspondence produces some blurriness within the closed boundaries as shown in Figure 5 (row with red outlines), which demonstrates that region correspondence helps refine closed areas. Overall, the absence of region-level correspondence also leads to a decreased performance compared with the full SAIN.

## 4.5 Sampling for Stroke Correspondence

The stroke correspondence $\mathbf{P}_t$ is continuous in terms of $t$. An ideal temporal modelling strategy needs to incorporate sufficient information without causing issues due to redundant input patterns. We investigated a number of settings as shown in Table 3. It can be observed $\mathbf{P}_t$ with $t \in \{\frac{1}{2}\}$ achieves the best performance. Conversely, providing more information may lead to an over-fitting issue, especially for the case with $t \in \{\frac{1}{4}, \frac{1}{2}, \frac{3}{4}\}$. As expected, only with the guidance of the middle temporal point $t \in \{\frac{1}{2}\}$ works the worst among all settings. However, with three sampling slices, where $t \in \{\frac{1}{4}, \frac{1}{2}, \frac{3}{4}\}$, the performance is worse than all two-slice based

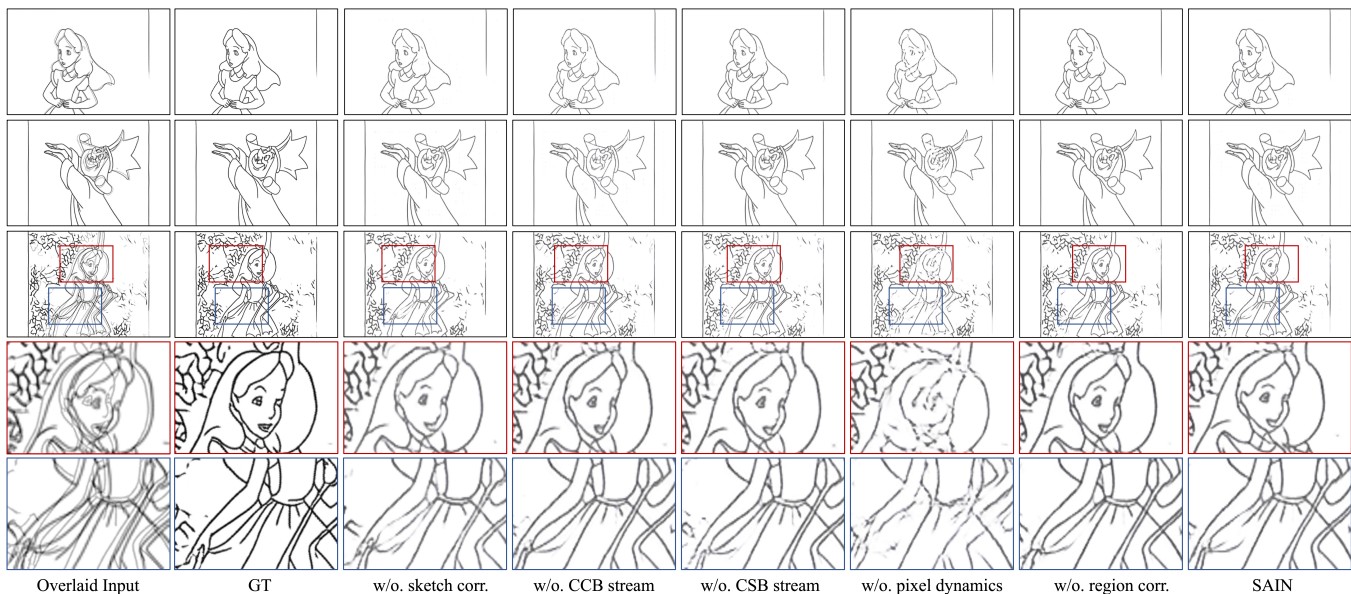

Figure 5: Qualitative samples of ablation studies with different components in SAIN.

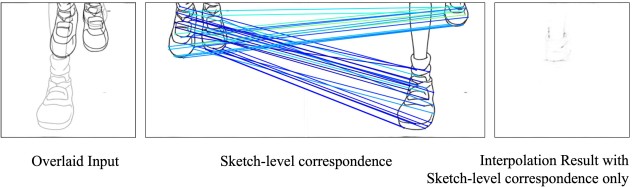

Figure 6: An example with extremely exaggerated motions.

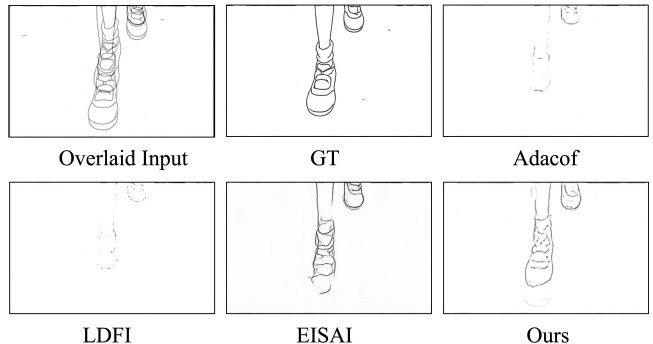

Figure 7: Interpolation outcomes of different approaches for the scenario with significant motions.

strategies, which suggest an over-fitting issue. For two-slice strategies, the most effective strategy is with the early motion patterns in $t = \frac{1}{4}$, highlighting the importance of the initial states.

## 4.6 Limitations & Future Work

Our proposed SAIN relies on the result of stroke and regional correspondence, which leads to limitations with the current correspondence mechanism. First, a linear stroke-level correspondence and a pre-defined temporal sampling strategy may result in less accurate interpolations. The future work should address this stroke-level correspondence scheme with a learnable manner based on the input sketches. Second, improvement in the scenarios of exaggerated motions is expected. In Figure 6, it can be observed that the strokes change extremely between the frames. As a result, the stroke keypoints are often with less confidence or incorrect for the point-wise matching and downstream modelling.

We present interpolation results for two frames featuring significant motions, as illustrated in Figure 7, alongside comparisons with existing Sketch (LDFI), Animation (EISAI) and Video (Adacof) interpolation state-of-the-art methods. Notably, Adacof and LDFI were unable to accurately capture the images within such a dynamic scene, resulting in blank outputs. EISAI, on the other hand, compromised the detail of the foot, leading to distortion, while our method successfully preserved the overall structure of the input.

## 5 CONCLUSION

In this paper, a novel deep learning method, namely, SAIN, is presented for animation sketch interpolation. Particularly, region-, stroke-, and pixel-level patterns are explored to take the sparse nature of sketch frames into account for interpolation. A multi-stream U-Transformer architecture is further devised to utilise the multi-level guidance with CSB and CCB. In order to evaluate our proposed method, a large-scale sketch dataset extracted from wild animation STD-12K was constructed for the first time. Comprehensive experiments clearly demonstrate the effectiveness of SAIN against the-state-of-the-art.

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
