# OpenReview forum: "Bridging the Gap: Sketch-Aware Interpolation Network for High-Quality Animation Sketch Inbetweening"
_acmmm.org/ACMMM/2024/Conference — MM2024 Poster_

### Official Review · Reviewer_Lx4d · 2024-05-17

**Rating:** 3
**Confidence:** 3

**Summary:**

This work proposed a sketch-aware interpolation network for animation sketch inbetweening. Given the sketching keyframes, the approach firstly computes multi-level correspondences, i.e. pixel-level dynamics, stroke-level correspondence and region-level correspondence. Then guided by these correspondences, a multi-stream U-transformer is devised to generate the inbetweening frames with CSB and CCB. Moreover, this work constructed a sketch dataset termed STD-12K which comprises 30 sketch animation series.

**Strengths:**

1.	This work investigates pixel, stroke, and region-level guidance to accurately comprehend the sparse characteristics of sketch animations.
2.	Extensive experiments have demonstrated the effectiveness of the proposed approach.

**Limitations:**

1．	This work lacks an important ref [1], which also converts open-source movies to full scenes in 2D styles including 2D Animation frames and contour lines. It provides ground-truth correspondences at pixel-wise and region-wise levels, respectively, where the pixel-wise labels are dense optical flows of cartoon sequences, and the region-wise ones are the matching of segments in adjacent frames. Many evaluation metrics have also been provided.

2．	Compared to reference [1], it is evident that this article lacks novelty. Firstly, all the modules proposed in this work, including pixel-level dynamics, stroke-level and region-level correspondences, transformer architecture, and the dataset, have been exploited in the field. Secondly, reference [1] provided ground-truth correspondences at pixel-wise and region-wise levels, along with corresponding evaluation metrics like EPE, occ, line, and flat, which are not included in this work. Thirdly, this work fails to evaluate the accuracy of correspondences at multiple levels. Additionally, the evaluation results in Tables 1 and 2 do not reach the SOTA compared to reference [1].

3.	Some limitations have been discussed in the limitations section of this article, such as exaggerated and large amplitude movements, and inaccurate stroke-level correspondence, which limit the practicality of this method.

4.	The proposed dataset STD-12K is extracted from 30 animation movies. Apparently, there is a significant bias between these data and hand-drawn sketches. Considering that the first sentence of this article emphasizes hand drawing, I believe the author needs to discuss hand-drawn sketches more, such as noise in keyframes and semantic confusion.

Ref:
[1] AnimeRun: 2D Animation Visual Correspondence from Open Source 3D Movies. Siyao, Li and Li, Yuhang and Li, Bo and Dong, Chao and Liu, Ziwei and Loy, Chen Change,  NeurIPS 2022.

**Suitability:**

2

---

### Official Review · Reviewer_CoPW · 2024-05-19

**Rating:** 3
**Confidence:** 3

**Summary:**

The work introduces a new method called SAIN for the sketch interpolation task. It features a novel design that integrates multi-level sketch-related guidance with U-Transformers. Additionally, a new large-scale dataset is proposed.

**Strengths:**

The paper introduces a new dataset that could potentially contribute to the development of the field. Additionally, the work achieves excellent results on the proposed dataset and provides a detailed experimental analysis. The comparison of this work with other baselines highlights its strengths and weaknesses, offering valuable insights for future research.

**Limitations:**

I apologize for any misunderstanding, but after carefully reading the paper, I still have the following questions:

1. I do not clearly understand the necessity of proposing this new dataset. Methods such as AnimeInbet also provide datasets, so why not compare your work on these existing datasets? Additionally, I noticed that the performance of the AnimeInbet method in Table 1 is quite poor, but there is a lack of analysis regarding this.

2. I do not fully grasp the advantages of your proposed dataset, nor the relationship between your dataset and existing related datasets. Is there an evaluation of your method's performance on other datasets?

I might have overlooked certain parts of the paper that address these points. If you could clarify these aspects, I would greatly appreciate it and improve the score.

**Suitability:**

3

---

### Official Review · Reviewer_TzWh · 2024-05-25

**Rating:** 4
**Confidence:** 2

**Summary:**

The authors propose a hand-drawn 2D in-betweening pipeline that utilizes motion dynamics, stroke-level correspondence, and region-level correspondence as inputs. Although more details on how these inputs are calculated are needed, the contribution seems adequate with the introduction of a new dataset.

**Strengths:**

1. The challenges of limited texture details and exaggerated changes between frames are well-identified, and the model design effectively addresses these challenges by constructing three correspondences and dynamics. Although it is unclear the detailed calculation of those correspondences and whether these correspondences are proposed for the first time, building such dynamics is pretty hard for sketch videos and does well represent the motion inside sketch animation.
2. A large-scale dataset, STD-12k, is proposed and claimed to be publicly available.
3. The Sketch-Aware Interpolation model fuses three correspondences and dynamics and appears novel and promising with detailed ablation studies.

**Limitations:**

1. Can this model be extended to handle freestyle, poorly-drawn sketches? It seems that the sketches used are derived from edge detection.
2. More details are needed on how optical flow is estimated between frames. It is unclear what model is involved or how the optical flow is calculated.
3. The abstract briefly mentions color details (#14), but there is no further clarification on what these color details are and how they relate to the grayscale sketches.
4. Some typos need to be carefully checked like inbewteening in #20

**Suitability:**

3

---

### Official Review · Reviewer_HimJ · 2024-05-25

**Rating:** 5
**Confidence:** 3

**Summary:**

The paper presents a new method - Sketch-Aware Interpolation Network (SAIN) for high-quality sketch animation inbetweening, which is an important yet challenging task with a wide spectrum of applications. A large-scale dataset has also been developed to be used in this work and further relevant studies. Extensive experiment results demonstrate the effectiveness of the proposed method.

**Strengths:**

Overall, the paper is well organised and easy to follow, and has made solid technical contribution to the sketch animation community. The newly developed dataset will also facilitate subsequent studies in this area.

**Limitations:**

The authors have provided a group of animations obtained with different methods in the supplementary demo video. They are encouraged to provide more examples in the demo video, for instance, all the instances that appear in the manuscript, and this will be helpful in evaluating the overall visual quality of the proposed method.

In addition, it would be interesting to see a comparison between different methods in handling sketches with varied details. For example, detailed sketches to highly abstractive sketches.

**Suitability:**

2

---

### Meta-Review · Area_Chair_hacB · 2024-06-27

**Recommendation:** Accept (Poster)
**Confidence:** 4

**Metareview:**

This paper was reviewed by four experts in the field. The paper received postive reviews WA, BA, BA, BA.
Reviewers like the new large-scale dataset, but raise the concerns of missing reference, novelty and bias of the dataset, etc.
After rebuttal, the reviewers come to a consensus that the concerns are well solved.
Considering the reviewers’ concerns, we are happy to recommend the acceptance of this paper.

---

### Meta-Review · Senior_Area_Chairs · 2024-07-10

**Recommendation:** Accept (Poster)
**Confidence:** 4

**Metareview:**

All the reviewers gave positive ratings and tend to accept the paper. SAC and AC agree with reviewers and recommend accptance of the paper.